# Patient Experiences of Communication with Healthcare Professionals on Their Healthcare Management around Chronic Respiratory Diseases

**DOI:** 10.3390/healthcare11152171

**Published:** 2023-07-31

**Authors:** Xiubin Zhang, Sara C. Buttery, Kamil Sterniczuk, Alex Brownrigg, Erika Kennington, Jennifer K. Quint

**Affiliations:** 1National Heart and Lung Institute, Imperial College London, London SW7 2BX, UK; xiubin.zhang@imperial.ac.uk (X.Z.); s.buttery@imperial.ac.uk (S.C.B.); 2Independent Researcher, London W12 0BZ, UK; 3Asthma + Lung UK, London E1 8AA, UK; ekennington@asthmaandlung.org.uk

**Keywords:** experience, communication, chronic respiratory disease, healthcare professionals

## Abstract

Background: Communication is an important clinical tool for the prevention and control of diseases, to advise and inform patients and the public, providing them with essential knowledge regarding healthcare and disease management. This study explored the experience of communication between healthcare professionals (HCPs) and people with long-term lung conditions, from the patient perspective. Methods: This qualitative study analyzed the experience of people with chronic lung disease, recruited via Asthma & Lung UK (A&LUK) and COPD research databases. A&LUK invited people who had expressed a desire to be involved in research associated with their condition via their Expert Patient Panel and associated patients’ groups. Two focus group interviews (12 participants) and one individual interview (1 participant) were conducted. Thematic analysis was used for data analysis. Results: Two main themes were identified and we named them ‘involving communication’ and ‘communication needs to be improved. ‘They included seven subthemes: community-led support increased the patients’ social interaction with peers; allied-HCP-led support increased patients’ satisfaction; disliking being repeatedly asked the same basic information; feeling communication was unengaging, lacking personal specifics and the use of medical terminology and jargon. Conclusions: The study has identified what most matters in the process of communication with HCPs in people with long-term respiratory diseases of their healthcare management. The findings of the study can be used to improve the patient–healthcare professional relationship and facilitate a better communication flow in long-term healthcare management.

## 1. Introduction

Chronic respiratory diseases remain a leading cause of death and disability globally, with asthma, chronic obstructive pulmonary disease (COPD), bronchiectasis, and interstitial lung disease (ILD) as the most widespread chronic respiratory diseases [1,2,3]. Globally, there were 544 million people living with a chronic respiratory disease in 2017, of whom 299 million had a diagnosis of COPD [4]. In the United Kingdom (UK), there are approximately 12.7 million people who have a history of asthma, COPD, or another longstanding respiratory illness [5].

People with long-term health conditions play a key role in their own care. Previous studies have indicated that for people with chronic diseases, there is a significant association between healthy lifestyle and health outcomes [6]. For example, in a multicohort study with 116,043 participants, it was reported that healthy lifestyle had the potential to provide disease-free years for patients with coronary heart disease, stroke, cancer, asthma, and COPD [6]. Additionally, studies of people with COPD indicate that several changes in lifestyle such as quitting smoking, appropriate physical exercise, and eating a healthy diet all impact health outcomes [7,8]. As lifestyle plays a substantial role in improving health, this requires clear, follow-up, monitoring, and support between healthcare staff and patients. In addition, for patients with long-term lung conditions, their perception of health-related risk factor management, health beliefs, and health behaviors play an important role in their health management. This can be influenced by the way health providers, family, and society communicate with them. For example, a clinician could say to a patient, ‘If you don’t stop smoking now, in 5 years’ time you’ll be too breathless to play with your grandchildren,’ instead of saying, ‘You will have a 10% chance of having a heart attack in the next year.’ This may affect a patient’s follow-through with recommended healthcare plans, which in turn either improves a particular health outcome or worsens it. However, how different styles of communication affect patients’ decision making on their health-related risk factor management has not been extensively studied from the patient perspective. One previous study found that coaching and tailoring communication style could increase patient satisfaction [9], while another study indicated that patient-centered communication is fundamental to ensuring optimal health outcomes [10].

In clinical settings, risk communication refers to dialogue surrounding health-related risk factors, between health professionals and patients, in order to make optimal healthcare decisions [11]. It is an essential clinical skill for the prevention and control of diseases because it can advise and inform patients regarding health management [12]. Long-term lung diseases require long-term care and drug treatment, necessitating frequent communication with healthcare professionals (HCPs) to facilitate recovery. So, it is very important to understand patients’ experience of how communication with HCPs affects their disease management.

This study aimed to understand the experience of communication between people with a long-term lung condition and their HCPs (a provider of health-care treatment and advice based on formal training and experience). The specific objectives were (1) to understand perceived experiences of conversation with HCPs on health-related risk information among people with chronic respiratory conditions; (2) to explore the barriers and challenges of effective communication between patients and HCPs about patient involvement in their healthcare management; (3) to investigate what approaches could facilitate better communication; and (4) to understand what patients deem good and/or poor qualities of communication.

## 2. Methods

### 2.1. Study Design, Participant Recruitment, and Data Collection

This qualitative study recruited people with chronic lung disease via Asthma & Lung UK (A&LUK) who advertised the opportunity to their Respiratory Voices Network, a group of 900 people who had expressed a desire to be involved in research associated with their condition and who receive a monthly bulletin of involvement opportunities by email. A research physiotherapist from our local Trust also helped with participant recruitment via the COPD research register. Participant information sheets were sent via email to potential participants who expressed an interest in the study and met the inclusion criteria. Written informed consent was obtained before conducting the interviews. People were eligible for inclusion if they had been diagnosed with a chronic lung disease and were aged 18 years and older. They were excluded if they were unable to give informed consent or were not fluent in the English language.

The interview questions were developed through a series of conversations with experts in the field (including experts from the research team, both clinical and lay members, A&LUK, and the A&LUK Expert Patient Panel). The feedback was used to inform revisions of the focus group questions and determine the total time taken to complete each focus group session. The questions were open-ended and covered introductory questions, exploratory questions, and specific health-related questions (Appendix B). The interviews were conducted via Teams due to the participants’ choice, audio-recorded with permission, and lasted approximately 1 h each; the minimum was about 52 min and the maximum was 75 min. Two researchers conducted the interviews together to ensure the interview questions were appropriately asked and answered the research question. Focus group interviews were set up with a flexible, unstructured dialogue to elicit the given topic. This method gives the opportunity to gain multiple perspectives through open discussion, and to reach data saturation [13]. Data collection was stopped when no additional new information appeared. Field notes were mainly used to record prompt questions during the interviews. All interviews were transcribed verbatim and imported into NVivo (2020 Release) for data management and data analysis.

### 2.2. Data Analysis

The data obtained from the focus groups and in-depth interview were combined during the analysis process, to corroborate and confirm the findings. Using the thematic analysis approach [14], the analysis process was undertaken using the following steps: (a) getting familiar with the data; (b) generating initial codes; (c) searching for themes; (d) reviewing potential themes; (e) defining and naming themes; and (f) producing the report. All themes and codes were checked by revisiting the transcript to ensure that the emerging themes remained grounded in the participants’ perspectives. Any conflicting elements on the interpretation and analysis of the data were discussed until an agreement was reached with the team. All team members reviewed the final subthemes and themes.

To ensure the validation and reliability of the study, we provided a clear and rigorous description of all the methodological steps used in the research process, from the research question and participant recruitment to collection and data analysis. One experienced qualitative researcher coded and analyzed the data independently, and the second researcher reviewed it subsequently to ensure accuracy and to reduce bias. To increase accuracy and consistency in communicating the results to the reader, we provided the adequacy of the original data in the report and followed the consolidated criteria for reporting qualitative research (COREQ) check list (see the Appendix A in PDF). Two participants provided feedback on the reports, which was used to inform revisions of the findings.

## 3. Results

As the study focused on understanding patients’ perceived experience of communication with HCPs, rather than investigating how personal demographic information impacted their experiences, the participants’ personal demographic information, such as their age and gender, was not collected. There were a couple of reasons why the patients’ demographic information was not included in the report: one reason was the ethical committee of the study organization was not willing to let us collect the interviewees’ demographic data, due to ethical considerations; another reason was some patients did not want to provide their personal information and requested that it not be collected. All the participants were coming from London area. Twelve participants took part in two focus groups and one participant participated in an individual interview (due to the available participating time inconsistent with the group discussions). The two themes of the study were ‘involving communication’ and ‘communication needs to be improved,’ and the corresponding seven sub-themes are shown in Table 1.

### 3.1. Involving Communication

The study identified the positive experience of the participants both from the support they received from allied HCPs and, more broadly, community support groups.

#### 3.1.1. Community-Led Support Increased the Patients’ Social Interaction with Peers

Most of the participants reported positive experiences from the support of community support groups. For example:


*‘it’s a nice sociable way without any pressure to talk about your… you know that you’ve all got similar problems. Your sympathetic and they’re always right.’*

*(P10)*



*‘But very positive experience...pulmonary rehab…just carried on and it’s been great through COVID-19…So that’s a really positive group experience.’*

*(P6)*


These two extracts reflect that peer support in particular helped people to develop their own goals and create strategies for self-empowerment. By sharing their own lived experience and practical guidance, these are concrete steps toward building a self-healthcare management.

#### 3.1.2. Allied-HCP-Led Support Increased Patients’ Satisfaction

The participants also shared their experiences of allied-HCP (health professions that are distinct from medical and nursing)-led support groups. For example:


*‘Charing Cross team offer a park walk and there’s always one of the respiratory team there, so I was having a chat with XX (a person name) who’s one of the physios there on the walk as well.’*

*(P11)*



*‘I agree on pulmonary rehabilitation that, it’s a nice sociable way without any pressure to talk about your rehabilitation.’*

*(P10)*


In the study, the allied HCPs who led support groups and community support groups as extended roles seem to have a positive impact on patients’ outcomes and are an essential part of the communication platform. These roles should be more emphasized.

In addition, participants also disclosed positive communication experiences they had with their consultants. For example: 


*‘I was very fortunate, that a lot of the people I spoke with were consultants and very specialized people. And that was the one of the big things that instead of appointments being 10 min or five minutes (with GPs), appointments were half an hour and they were available by phone and various things. So, I was very fortunate, really.’*

*(P9)*


As an appointment time with a consultant is more likely to be longer than a GP appointment, patients have sufficient time to communicate, which was highlighted by the participants. This reflects that the length of time seems to play an important role in forming a good conversation.

### 3.2. Communication Needs to Be Improved

In the study, participants disclosed several pitfalls around conversations with their HCPs. We categorized the way in which communication was disliked (negative aspects of communication) into five sub-themes: (1) disliked being repeatedly asked the same basic information; (2) unengaged communication; (3) conversation lacked personal specifics; (4) medical terminologies used affected communication; and (5) lack of sufficient information.

#### 3.2.1. Disliked Being Repeatedly Asked the Same BASIC Information

Participants expressed that common questions were repeatedly asked, leaving them feeling that the conversation was just a waste of the appointment time. They reported that they would rather have HCPs discuss specific risk information that they did not routinely pay attention to in their daily life, such as 


*‘I always get asked about smoking... And yet nobody ever says to me. Do you have a wood stove? And of course, I wouldn’t dare. This was the problem. The neighbor’s wood stove...’*

*(P1)*



*‘And allergies and nothing else. Yes, and maybe road traffic? Maybe road traffic now, but nobody ever says you know where you live...’*

*(P1)*



*‘If you have to repeat somethings that you’ve told them before...’*

*(P11)*


From the above quotes, it could be seen that the participants disliked being repeatedly asked for the same basic information. This communication approach also brought a negative communication experience for them.

In addition, asking repeatedly about basic information led to emotional distress and the participants perceived the experience of being treated as a child. For example:


*‘Now I feel like I’m about 5 and I’m very stupid and you know …he seems to have the view that patients do not need to know this.’*

*(P2)*



*‘The only thing in risk management that I’ve been told over and over again, and considering the fact that after my first exacerbation in 2014, I haven’t done this…So how stupid do they actually think people are?’*

*(P5)*



*‘I don’t know what they put on their system. Maybe they don’t put everything that they supposed to. And then I have to kind of repeat myself all the time.’*

*(P13)*


Being constantly asked the same basic information made the participants feel foolish and may have damaged their relationship with their healthcare professionals. Therefore, with chronic disease patients, avoiding repeatedly asking basic information and having clinical conversations that tend to provide more specific information about their diseases might improve the patients’ involvement with disease management and adherence to a long-term healthcare plan.

#### 3.2.2. Unengaged Communication

Individuals felt excluded from their treatment-decision-making process and healthcare management, feeling unengaged and unempowered when communicating with their HCPs. This made the participants feel that they had been given a lecture, rather than undertaking a dialogue.


*‘...it’s quite sad because how many patients aren’t involved in those decisions in their care because they can’t articulate what they’re trying to say, or because they can’t interpret their results or their letters, or are empowered enough, because it hasn’t been explained to them.’*

*(P2)*


Another participant perceived the feeling of being treated as an object rather than a person. 


*‘I feel like they treat us as a number.’*

*(P13)*


Similarly, a conversation without a discussion made the participants feel that they were not involved in communication and created stress. For example, one participant said


*‘whatever your condition is, if they could immediately put that into perspective for you… it would reduce the anxiety of very significant amount...’*

*(P6)*


The below extract shows common themes of ‘not listening,’ ‘not enough time to communicate,’ and the important issue that non-verbal communication (such as nipping next door) plays an important part in the communication.


*‘… a 20-min appointment and then you get to the end of it. And it’s like ohh, I’ll just go and see what the consultant says, so they’ll nip next door and they’ll come back and say, ohh, we’ll just carry on with what we’re doing. See you in six months.’*

*(P3)*


Reassurance has been recommended in clinic settings; however, the participants reflected that reassurance without full discussion led to unclear communication. For example, one participant said


*‘I find it confusing because I think I understand for myself what my risk factors might be. But this is not something that I can discuss with my team, or if I do, it feels like there’s a heavy dose of reassurance going on, and I don’t necessarily feel that reassurance is anything, I feel it’s misplaced because I didn’t want reassurance. I just wanted to have an adult discussion about my health.’*

*(P2)*


It seems that adult discussion and clear explanations are important to empower patients in their healthcare decision making and form good communication and clinical instructions for patients.

#### 3.2.3. Conversation Lacked Personal Specifics

Participants also explained how there was a lack of personal, specific conversations with their HCPs. One participant said


*‘you know, this all comes down to the fact that not everybody’s the same. You know some people have different attitudes to things.’*

*(P11)*



*‘A bit tailored like I don’t need to be confronted with some with the risk cause I’m aware of it, but it will be helpful…’*

*(P2)*


Most of the participants think that a conversation that relates to ‘me’ matters more for them, giving the conversation a sense of it being personal and being immediately applicable. For example, the two extracts below expressed how participants thought about communication.


*‘So, it’s a really difficult thing to do because at its core it’s statistics and percentages and you know a lot of us don’t deal with that on daily basis and it can be quite difficult. So I often found it was. It was easier when they communicated in analogies, you know, metaphors or stories and that type of thing…’*

*(P9)*


Offering a tailored care approach and having options were also expected by the participants.


*‘I feel like the personal life one might be more likely… I want them to understand how it relates to me.’*

*(P2)*


It seems that the participants were expecting a personalized conversation about their disease management from their healthcare professional. Personalized communication may therefore help to form a better guide for individual risk management, as well as help the flow of communication between patients and HCPs.

#### 3.2.4. Medical Terminologies Used Affect Communication

Much evidence shows that the medical terminologies used in clinical settings affect communication with patients. In general, HCPs may be aware of avoiding the use of medical terminology when they are face to face with patients. But they may ignore the same issue in extended communication scenarios or situations, such as a reference letter delivered to the patient. In this study, the participants disclosed such situations.


*‘I can’t interpret…I spent three hours while translating my consultant’s letter and I literally write scientific papers. I still couldn’t… turned out like he was using a definition of asthma that literally no one else uses. So, I was very puzzled by it and I spent a while. And after a lot of digging through PubMed, I think I finally understood what he was on about.’*

*(P2)*


In another situation, the communication of clinical explanations to the patients also affected understanding about their health condition. An inappropriate clinical explanation causes emotional distress for the patients and possibly leads to a bad starting point of healthcare management. It also may lead to worse self-management by the patients if they wrongly interpreted what their HCPs said, just as one participant said


*‘You know, it’s immediate panic basically… they give you a percentage and things like that of your chance of death in the next five years. And how do you interpret that? And then it’s sort of well, what’s the next stages.’*

*(P9)*


Scientific statements may also make patients feel excluded from their communication with HCPs as well as feeling a loss of empowerment of their healthcare management. For example, one participant said


*‘I feel like every patient should have the right to be able to be involved in those conversations and it’s really sad that only at the point that you can almost prove to the consultant that you can engage in those conversations.’*

*(P2)*


Therefore, avoiding the use of scientific terminology when communicating with patients may help to create good communication, and to create an easy follow-up procedure of the clinical instructions of disease management for patients. This is perhaps less of a problem for an informed patient who is interested in his/her diseases and may be inclined if not understanding something to ask for an explanation or look it up later.

#### 3.2.5. Lack of Sufficient Information/Services

People felt that HCPs did not provide sufficient information to patients, which created tension and disappointment. One participant expressed feelings of the first time he heard about his condition from his GP. Unclear information made him puzzled and anxious.


*‘When you’re first told it’s a bit of a shock. The important part really would be if they could immediately put that into perspective for you, I think it would in my particular case, it would reduce the anxiety of very significant amount.’*

*(P5)*


A lack of integration between healthcare teams can act as a barrier to receiving continuity of care and affect the flow of communication. Here is an example given by a participant about her experience with the healthcare service that reflects the lack of integration between healthcare teams.


*‘When you’ve been seen by lots of different specialists...sometimes when you go to a different consultant, he tells you something different to what you’ve been told…’*

*(P4)*


The participants expressed a preference for continuity for follow-up to build relationships.


*‘...even just seeing the same person every time is a big thing because you build up a relationship with that person.’*

*(P3)*



*‘The only sad thing was there was really no follow up…I think it would have done a lot of good to a lot of people.’*

*(P7)*


Furthermore, other services being unavailable, difficulty in getting a timely appointment, or fewer face-to-face appointments because of COVID have also affected communication.

Wasting time searching a patients’ medical history may reduce the communication time as one participant commented


*‘I wondered whether the problem is that in my case my file must be huge. In the notes, I wonder whether they have a top page that actually just lists anything important, like the things you’re sensitive to, your allergies, or the most recent tests, or the fact that you’re a non-smoker. Cause I could almost feel that the consultant sort of going back wading through all these pages. I thought does he not have a summary page on top?’*

*(P1)*


Overall, as the participants discussed, there are many factors that affect the flow of communication between patients and HCPs during the treatment process and healthcare management. Communication was particularly felt to be an issue when care is delivered in two trusts who use different labs and IT systems, and patient records are not shared effectively. The importance of continuity of care, i.e., seeing the same HCPs, helps to avoid banal questions and make the most of appointments. Some doctors make an effort to familiarize themselves with the patient’s case, e.g., by reading the last consultant’s letter. Some GPs rush the patient through the 10 min appointment, while they type and answer the phone, and will not answer questions unrelated to the problem.

## 4. Discussion

This study highlighted participants’ perceived experiences of communication with their HCPs about their disease management. The main areas of poor communication raised centered around the pitfalls of single dialogue, and unclear explanations or a lack of specifics, etc., from the patients’ perspectives. Some responses were associated with service infrastructure issues while others were related to communication technique problems. By identifying specific pathways through which communication can lead to better understanding about health-related risk factors, we can provide a better communication approach to support patients with their healthcare management.

In our study, the participants described their emotional feeling of exclusion and hierarchy in the process of talking with their HCPs, which suggest that tailoring discussion to make it understandable but not patronizing is very important. The influence of psychological factors (mental health) due to physical health on managing lifestyle/improving quality of life should not be underestimated. There are many references to emotional distress and anxiety as a result of communication with HCPs. We found that the explanation and interpretation of messages are important issues in the communication process. The finding suggests that different interpretations of health information can elicit different responses from patients. Ineffective communication in clinical settings can prevent optimal healthcare decisions, making significant impacts on patients’ healthcare outcomes [15]. As such, an in-depth knowledge of their disease can be an effective way in increasing patients’ motivation to take correct self-care actions. Previous studies also evidence that individuals will be more likely perform effective self-management of their condition if they can be more engaged with the process, and being more informed and confident in their skills, knowledge, and abilities [16,17].

The link between miscommunication and poor patient outcomes has been well studied [18,19]. However, studies on how and why communication might impact patients’ health outcomes are limited. This requires a deeper understanding of how specific aspects of communication are linked to specific outcomes, and how contextual factors affect the understanding of communication on healthcare management. This is especially relevant for people with long-term health conditions and chronic diseases, who are less likely to be influenced by a single clinician–patient encounter, and more by the cumulative effect of the patient’s communication over time with their physicians. Appropriate words used for medical explanations in oral conversation have been explored by many previous studies [20,21]. We found in our study that medical terminologies used in patients’ referral letters can also affect clinic communications, something that is more likely to be ignored by healthcare professionals. Medical terms can be interpreted differently by patients without full understanding, which should be considered by HCPs when they work with patients from different scientific or cultural backgrounds. So, taking time to explain, allowing patients time to digest, and then returning with more questions are needed.

Poor communication or advice also prevents patients from medical adherence or results in patients being disappointed with their care [22]. Studies illustrated that ineffective communication may lead to an underestimation when communicating the severity of a patient’s illness, and the inappropriate management of health-related risk factors [23]. One recent study indicated that miscommunication between patients and HCPs can negatively impact medical care, patient quality of life, and both patients and HCPs’ engagement in shared decision making [24]. HCPs should provide options when it comes to clinical decision making; there may clearly be one option that is best from an HCP standpoint, but explaining to the patient why is paramount rather than just expecting them to accept it.

The person-centered model of care is widely applied in clinical settings [25,26,27]. It means that an HCP’s role is to support patients in being active agents regarding their healthcare management rather than being a sole expert telling patients what they should to do. Many previous studies indicate that person-centered care has empowered autonomy and the involvement of patients in their healthcare decision, and increased interactions between patients and HCPs [25,28]. Practically, it may be challenging when a patient presents a desire for healthcare management that differs from evidence-based recommendations for best practice [29]. As there are evidenced benefits to person-centered care within the context of chronic condition self-management [30], we should take into account the persons’ day-to-day experience, knowledge, and action over time in managing their health, to strengthen and improve a person’s control over their health. Sharing knowledge and decision making is also vital for facilitating patient-centered care [31], and improves the relationship between patients and HCPs.

The study also found that integrated services are very important for the continuity of care that participants received. They reported a familiarity of their HCPs, which also saved them having to repeat the entire history of their condition on each visit. Previous studies reported that integrated care links multiple levels of care services and makes it easy to communicate for HCPs, makes medical decision making faster, and also supports a continuum of care and better health outcomes while controlling costs [32,33,34]. In our study, the participants also valued the support they received from the allied HCPs and community groups, which helped them to establish a good self-care management plan. The peer support service has also provided an opportunity for the participants to share their experience of living with long-term health conditions and skills of self-management. In addition, the participants also claim to have more positive communication experiences with consultants as a consultant appointment provides more specific information with longer time than a GP appointment. As multi-disciplinary team is recommended in all working settings, especially in patients’ self-healthcare management of chronic disease [35,36]. Therefore, the availability of such services and increasing patients’ awareness of services provided by the hospital or a charity organization is important. It also suggests that integrated care teams may include awareness of services provided by hospitals at home or virtual respiratory wards, and providing opportunities for peer support/reassurance/common ground to ease psychological worries is important for a balanced health management for the patient.

There are a couple of limitations of this study that should be recognized. Firstly, the study did not collect the participants’ personal information (such as age and gender), which might limit the generalizability of our results. However, this might not be a significant issue, because the study focuses on exploring and interpreting the meaning of communication for people with long-term lung conditions, rather than to analyze how the sociodemographic characteristics impact their experiences of clinical communications. Secondly, all participants were recruited from inside London, which may limit the heterogeneity of the sampling. Thirdly, the study may be biased by people who commonly only tend to focus and remember the negative and, therefore, there is some self-selection around who ultimately may have decided to take part in this study. We are also aware that the study findings lack specificity with chronic respiratory diseases; however, our study aims to understand the patients’ communication experience with their healthcare professional. Therefore, it is not surprising that the findings/results are commonly about communication issues rather than specific to the patient’s types of diseases. However, the study findings not only explored the personal communication experiences of patients with chronic respiratory disease with their healthcare professional, but also raised the question, ‘how communication adds to the chronic patients’ involvement on their disease and healthcare management.’

## 5. Conclusions

This qualitative study has identified what most matters in the process of communication with HCPs in people with long-term respiratory diseases and preferences of communication from the service users’ perspectives. For people with chronic respiratory diseases, effective patient care is not ‘telling patients what to do,’ but supporting the service user and upholding personal needs and patient-involved care management. The findings of the study can be used to develop a long-term collaborative patient–clinician relationship and healthcare management via better communication flow.

## 6. Patient and Public Involvement

PPI led and two patient representatives were helped with the development of interview questions and recruitment. One of A + LUK’s Expert Patient Panels—groups of 12 people with a range of lung conditions who are highly engaged in both their condition and research and meet every 2 months to discuss a range of topics—reviewed the topic guide in March 2022 and provided feedback throughout the document. This feedback informed a number of changes to the topic guide, and the research team also provided responses back to the Expert Patient Panel to show where their input had resulted in a change.

## Figures and Tables

**Table 1 healthcare-11-02171-t001:** Themes and sub-themes of the study.

Themes	Sub-Themes	Quote Examples
**Communication needs to be improved**	Disliked being repeatedly asked the same basic information	*‘And allergies and nothing else. Yes, and maybe road traffic? Maybe road traffic now, but nobody ever says you know where you live...’ (P1)* *‘Now I feel like I’m about 5 and I’m very stupid and you know …he seems to have the view that patients do not need to know this.’ (P2)* *‘If you have to repeat somethings that you’ve told them before...’ (P11)*
	Unengaged communication	*‘...it’s quite sad because how many patients aren’t involved in those decisions in their care because they can’t articulate what they’re trying to say, or because they can’t interpret their results or their letters, or are empowered enough, because it hasn’t been explained to them.’ (P2)* *‘I feel like they treat us as a number.’ (P13)*
	Conversation lacked personal specifics	*‘You know, this all comes down to the fact that not everybody’s the same. You know some people have different attitudes to things.’ (P11)* *‘I feel like the personal life one might be more likely… I want them to understand how it relates to me.’ (P2)*
	Medical terminologies used affected communication	*‘I can’t interpret...I spent three hours while translating my consultant’s letter and I literally write scientific papers. I still couldn’t… turned out like he was using a definition of asthma that literally no one else uses. So, I was very puzzled by it and I spent a while. And after a lot of digging through PubMed, I think I finally understood what he was on about.’ (P2)* *‘You know, it’s immediate panic basically… they give you a percentage and things like that of your chance of death in the next five years. And how do you interpret that? And then it’s sort of well, what’s the next stages...’ (P9)*
	Lack of sufficient information	*‘…The important part really would be if they could immediately put that into perspective for you, I think it would in my particular case, it would reduce the anxiety of very significant amount.’ (P5)* *‘When you’ve been seen by lots of different specialists...sometimes when you go to a different consultant, he tells you something different to what you’ve been told…’ (P4)*
**Involving communication**	Community-led support increased the patients’ social interaction with peers	*‘it’s a nice sociable way without any pressure to talk about your… you know that you’ve all got similar problems. Your sympathetic and they’re always right.’ (P10)*
	Allied-HCP-led support increased patients’ satisfaction	*‘Charing Cross team offer a park walk and there’s always one of the respiratory team there, so I was having a chat with XX (a person name) who’s one of the physios there on the walk as well.’ (P11)*

## Data Availability

Data are available on reasonable request.

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
