# Peer review of "Patient Experiences of Communication with Healthcare Professionals on Their Healthcare Management around Chronic Respiratory Diseases"

_healthcare, 2023, doi:10.3390/healthcare11152171_

Round 1

Reviewer 1 Report

Healthcare provider communication is the key to engagement and patient outcomes-- thank you for highlighting this aspect of care and the importance.

Your papers offers a big focus on the negative aspects of participants in the table and in the results but a no discussion of the positive aspects in the table and limited discussion about person-centered care, how it was perceived and if HCP's can improve on it, especially in this population.  Can you explain why you chose the patients for the project? Do they have worse health outcomes?  Have you noticed that your patients are unhappy or not following recommendations? What prompted you to do conduct this project and do you believe it is applicable to other patients with chronic diseases?  Only 12 participants makes it hard to generalize the results.

Why do you think there are so many Negative themes and only one positive?  This is concerning.

The most important finding is the last two paragraphs (Line 414- forward) but you haven't suggested how we need to improve on this or decrease the negative aspects of communication that is suggested in your findings.

Several of the references are more than 10 years old-- please update or eliminate them.

Minor grammatical errors (punctuation mostly) throughout that could be easily corrected with editor.

Thank you for the opportunity to read your submission and for your efforts to improve patient outcomes.

Minor punctuation errors-- mostly use of commas that are not needed. 

Author Response

Dear Reviewer1,

We greatly appreciate the comments you provided, which have helped us to improve the quality of the paper. We responded to all your comments point-by-point, and indicated where changes to the manuscript have been made (e.g. page, line).  We also highlighted the changes to the manuscript with track changes.

Reviewer 2 Report

Thank you for the opportunity to review this manuscript. The topic is very relevant.

General comments:

It is difficult to get hold of what the aim of the study is, as different aims are reported.

The terms “communication”, “decision making”, and “shared decision making” must be clarified and used consistently.

Braun & Clarke describes an inductive analysis of data; however, it seems like the study uses a deductive analysis: You want to explore positive and negative experiences and find themes that are named the same.

More specific comments are listed below.

Abstract

Line 19: There are two types of characters.

It would improve the manuscript if the aim was more focused on a topic than on “positive and negative” experiences, for example communication about disease management (which is described at line 361-362), communication about prevention and disease control or communication about patient involvement in decisions.

Line 28-30: The “conclusion” is a perspective and not a conclusion.

Manuscript

Line 53-60: Lack of reference.

Line 60-62: You claim that different styles of communication has not been investigated extensively, which means that some studies have been conducted. What did the studies investigate and find?

It is not clear what the focus of the study is. Does the study focus on communication in general, on patient decision making or shared decision making (the study title)?

Line 74-79: Here is a different aim than in the abstract.

Line 95: I guess the interview question were related to the aim of the study?

Line 133 – 135: Demographic data on the participants are need, otherwise, the results of the study cannot be compared with other studies or transferred to other clinical settings.

Line 138-140 + table 1: Negative and positive aspects of communication cannot be themes.

The themes and thereby the result section must be revised. The result section includes many quotes, less quotes and more interpretation are recommendable.

Author Response

 Dear Reviewer2,

We greatly appreciate the comments you provided, which have helped us to improve the quality of the paper. We responded to all your comments point-by-point, and indicated where changes to the manuscript have been made (e.g. page, line).  We also highlighted the changes to the manuscript with track changes.

Reviewer 3 Report

The study addresses a relevant issue and has explored this through participatory research. However, there are some aspects that make it difficult to classify and interpret the results appropriately. Please see my corresponding comments and suggestions below.

General

The title of the paper includes shared decision-making. However, this concept is neither mentioned nor explained or used throughout the manuscript. Therefore, the term should not be used in the title.

In several paragraphs throughout the manuscript, there is one blank space too much. Please revise/correct.

Method

2.1 Study design, participant recruitment and data collection

p. 2, line 83: Although the abbreviation “A&LUK” has already been explained in the abstract, it would be helpful to do so again here; non-UK readers are probably not familiar with it.

p. 2, line 89: Please write “interest” instead of “interesting”.

p. 3, line 100/101: What was the rationale for conducting the interviews online? And were both focus groups and individual interviews conducted via Teams?

p. 3, lines 101/102: What was the minimum/maximum duration (i.e., range) of the interviews?

p. 3, line 104: Please write “interviews” instead of “interview”.

Line 106: Please delete the “PhD” following “Fusch”. 

2.2 Data analysis

p. 2, line 120: How many team members were involved in the review?

Results

p. 3, lines 133 et seq.: The rationale given for why demographic information on participants were not collected is not clear to me. This is important additional information that enables an appropriate evaluation and interpretation of the study results and should generally be independent of the nature of the research question. As it stands, the results are isolated, so to speak, and without reference points to other research.

pp. 3/4, table 1: It’s a bit odd that only some of the categories/themes are illustrated by quotes while others are not. Please consider renaming the table with a reference to “negative aspects of communication” or add in the respective cells that the subthemes of “positive aspects…” and sample quotes are outlined in the subsequent paragraphs.

In addition, the table might be placed as a summary at the end of the results section.

p. 4, line 144, as well as p. 2, line 74: It would be helpful to briefly explain what “allied health care professionals” are, as several readers are probably not familiar with it (which professions does the term encompass etc.).

p. 4, line 163 et seq.: Please italicize the quote, analogous to the other quotes.

p. 5, line 180: Please add “to” in the sentence (“…length of time seems to play an important role…”).

p. 5, line 194: Please add “have” (“…they would rather have HCP discuss…”).

p. 5, lines 214 et seq.: This sentence is a bit difficult to understand. Moreover, I would suggest placing it in the discussion section where it seems more appropriate.

p. 8, line 350: Please add “that” (“…there are many factors that affect…”).

p. 8, line 355: Please correct “8amiliarize”. 

Discussion

p. 9, line 414: Please add “is” (“The person-centered model of care is widely applied…”).

In this context, the concept of shared decision-making mentioned in the article title could be discussed as it can be considered an important component of patient-centered care. Barriers to SDM implementation could also be discussed in this context, bridging to some of the difficulties and experiences reported by participants in this study. See for example Tang et al. (2022) (DOI: 10.1111/hex.13541) or Truglio-Londrigan et al. (2014) (doi:10.11124/jbisrir-2014-1414).

p. 10, line 436: Please correct “aawareness”.

p. 10, lines 440 et seq.: Another limitation I see is the lack of specificity in the results – to what extent are the experiences reported by participants typical of individuals with chronic respiratory diseases? What can be derived from the results as recommendations for healthcare and disease management for this specific patient group? There is no information on this in the results or discussion sections.

As mentioned above, I find the rationale for not collecting sociodemographic data not convincing.

Author Response

Dear Reviewer3,

We greatly appreciate the comments you provided, which have helped us to improve the quality of the paper. We responded to all your comments point-by-point, and indicated where changes to the manuscript have been made (e.g. page, line).  We also highlighted the changes to the manuscript with track changes.

Round 2

Reviewer 3 Report

Dear authors,

thank you for revising your manuscript and considering my suggestions and comments. In conclusion, I would suggest that the following issues you raised in your cover letter be addressed in the manuscript prior to publication:

Please add the information on minimum/maximum length of interviews in the paper (methods section, 2.1 study design....).

Please add the information on why sociodemographic data was not collected in the results section.

Please explain HCPs in the introduction when outlining the research questions of the study (lines 81 et seq.), rather than in the results section.

line 485: Please delete "are" in the sentence, i.e., "However, the study findings not only explored..."

line 509: Please write "participants", not "participant".

Author Response

Dear Reviewer 3,

We greatly appreciate the comments you provided, which have helped us to improve the quality of the paper. We responded to all your comments point-by-point, and indicated where changes to the manuscript have been made (e.g. page, line). We also highlighted the changes to the manuscript with track changes.

Thank you again for your reviews.

Best wishes,

Xiubin Zhang

Comments and Suggestions for Authors

1. Please add the information on minimum/maximum length of interviews in the paper (methods section, 2.1 study design....).

Authors’ response: Many thanks for your comments, we have added the information on minimum/maximum length of interviews in the paper. Please see the page 3, line 107-108.

2. Please add the information on why sociodemographic data was not collected in the results section.

Authors’ response: Many thanks for your comments, we have added information on why sociodemographic data was not collected in the results section. Please see page 3, line 142-146.

3. Please explain HCPs in the introduction when outlining the research questions of the study (lines 81 et seq.), rather than in the results section.

Authors’ response: Many thanks for your comments, we have  explained HCPs in the introduction section, please see page 2, line 77.

4. line 485: Please delete "are" in the sentence, i.e., "However, the study findings not only explored..."

Authors’ response: Many thanks for your comments, we have deleted "are" in the sentence, please see page 10, line 469.

5. line 509: Please write "participants", not "participant".

Authors’ response: Many thanks for your comments, we have changed "participant" to "participants", please see page11, line 492.